## Case Study

coastal social-ecological systems; climate change; community adaptation; indigenous and local peoples; Ghana

**Corresponding author:**
Eranga K. Galappaththi;
Email: eranga.research@gmail.com

# Voices from Akplabanya: Community adaptation and social-ecological changes in coastal Ghana

Eranga K. Galappaththi [ID], Brandy Ayesu-Danso [ID], Sithuni M. Jayasekara [ID], Timothy D. Baird [ID], Anamaria Bukvic [ID] and Santosh Rijal [ID]

Department of Geography, Virginia Polytechnic Institute and State University, Blacksburg, VA, USA

## Abstract

Despite coastal regions' importance and vulnerability to climate change, Ghana's coasts remain underexplored through social-ecological systems (SES) approaches, with limited attention to Indigenous and local communities' adaptive responses to contemporary challenges. We conducted a study with the aims of (1) identifying the changes in coastal SES as perceived by the Akplabanya community and (2) examining the Akplabanya community's human adaptation responses to those changes. During two months of fieldwork in Akplabanya, we used four data collection methods: participant observation, semi-structured interviews, key informant interviews, and focus group discussions. We found social-ecological changes related to five themes: (i) coastal climate change (sea-level rise), (ii) resource change (changes in land use), (iii) agrobiodiversity loss (changes in livestock), (iv) pollution (unsustainable practices) and (v) population change (increasing population). As adaptation responses to these changes, the community adaptive responses we found were (a) place (sense of place), (b) agency (emergence of food markets), (c) Indigenous and local knowledge (weakening of Indigenous knowledge), (d) collective action (collective solutions), (e) institutions (partnerships) and (f) learning (awareness). Our study highlights the urgent need for targeted research in regions like Ghana to guide and improve adaptation policy interventions for scientists, policymakers and researchers.

## Impact Statement

This study offers critical insights into how coastal communities, like Akplabanya in Ghana, are experiencing and responding to environmental and social change. As coastal erosion, sea-level rise, biodiversity loss, and climate variability intensify, understanding localized responses becomes vital for developing effective adaptation strategies. Through in-depth fieldwork, this research highlights not only the environmental stressors affecting Akplabanya but also the community's strengths in adapting through collective action, use of new technologies, and a strong sense of place. The broader impact of this study lies in its contribution to global conversations about climate resilience and sustainability, particularly in marginalized communities that are often overlooked in policymaking. By framing adaptation within a social-ecological systems (SES) approach, the research underscores the importance of integrating Indigenous knowledge, institutional partnerships and community-driven innovations into national and international climate adaptation planning. Policymakers, Non-governmental organizations (NGOs) and researchers can draw lessons from Akplabanya's experience to inform climate adaptation policies in other vulnerable coastal regions. For example, strengthening local institutions, supporting culturally grounded adaptation strategies and preventing maladaptation – like environmentally harmful fishing practices – can lead to more sustainable outcomes. This work also calls attention to the urgent need to document and preserve Indigenous knowledge systems, which are rapidly eroding but remain essential for community resilience. Ultimately, this study helps bridge the gap between high-level climate policy and the lived realities of coastal communities on the frontlines of environmental change.

## Introduction

The ocean covers about 70% of the Earth's surface (McLean et al., 2001). Over 500 million people reside in coastal areas (Hossain et al., 2020). The coast is one of the most diversified ecosystems in the world, serving various important functions such as providing recreation and tourism opportunities, acting as a medium for transportation, serving as a repository of genetic and biological information, and functioning as a sink for waste (Harley et al., 2006; Cisneros-Montemayor and Ota, 2016). Coastal areas are particularly important for Indigenous peoples, as coastal Indigenous peoples consume ~2.1 million metric tons of seafood annually, accounting

for around 2% of the global commercial fisheries catch (Cisneros-Montemayor et al., 2016). Indigenous peoples living near the ocean have vital links to marine ecosystems. For millennia, the ocean has shaped the cultural heritage and spiritual values of coastal Indigenous peoples while providing essential food and economic security (Cisneros-Montemayor and Ota, 2016).

Coastal communities face significant environmental and social changes. Climate change, in particular, has marked impacts on these coastal communities (Adger et al., 2005). According to the IPCC (2008), coastal and marine environments are highly vulnerable and especially likely to be affected by climate change (Bernstein et al., 2008; Moreno and Amelung, 2009). The main climate change impacts include sea-level rise, coastal erosion, coastal flooding and changes in temperature (Nicholls and Cazenave, 2010; Potter, 2014). These changes have resulted in disruptions to livelihood activities, damage to infrastructure, and displacement of populations (Adger et al., 2005). In addition to environmental changes, coastal communities experience social changes, such as outmigration from coastal areas (Lawyer et al., 2023). This migration can be driven by the loss of homes and economic opportunities due to environmental degradation (Milán-García et al., 2021). Coastal Indigenous peoples are particularly vulnerable to these challenges because of their deep dependence on the ocean for cultural, spiritual and economic reasons (Cisneros-Montemayor and Ota, 2016).

Coastal ecosystems and social systems are deeply interconnected, and both have experienced significant climatic and nonclimatic changes over the past decades (Refulio-Coronado et al., 2021). These changes are best understood through the concept of coastal social-ecological systems (SES), which highlights the interdependence between human activities and environmental conditions in coastal areas (Berkes et al., 2003; Fischer et al., 2015; Hossain et al., 2020). This interdependence shows how the resilience of coastal systems is tied to the well-being and actions of local communities (Folke, 2006; Bernhardt and Leslie, 2013). Coastal SES studies emphasize the dynamic interactions between social and ecological factors, highlighting the risks that environmental and social changes pose to resilience (Adger et al., 2005; Olsson et al., 2007). Community perceptions of these changes are critical for determining the community's capacity to adapt; thus, the act of fostering resilience and sustainable resource management requires understanding these interactions (Adger et al., 2005; Bennett and Dearden, 2013).

Despite its importance, research on Ghana in the context of coastal SES is limited (Williams et al., 2020; Amadu et al., 2021). The available research does not adequately address the current effects of SES changes on Indigenous peoples or their adaptation responses (Ferro-Azcona et al., 2019; Freduah et al., 2019). For example, Williams et al. (2020) analyzed the costs and benefits of implementing climate adaptation strategies among Ghanaian smallholder horticultural farmers. However, because this research does not focus on Indigenous peoples, it leaves a gap in understanding how SES changes impact Indigenous communities. Physical changes, such as coastal erosion and inundation of Ghana's coastline, have been documented, but the interdependence between these impacts under climate change remains underexplored (Boateng, 2012; Evadzi et al., 2018; Arkhurst et al., 2023). Specifically, there is little research documenting ongoing coastal SES changes in the Indigenous coastal community of Akplabanya in the Ada West District (Cudjoe and Kwabla Alorvor, 2021; Dada et al., 2021; Loch and Riechers, 2021).

The Akplabanya is an Indigenous community that heavily relies on small-scale fisheries for livelihood and contributes significantly to Ghana's national economy. With approximately half of Ghana's 540-km coastline threatened by sea-level rise – especially in central and eastern coastal regions like Akplabanya (Boateng et al., 2017) – the community faces increasing exposure to coastal erosion, inundation and resource degradation. Yet, the existing adaptation strategies, including restrictions on fishing during holidays, have become less effective due to economic hardships and limited communication between fishers and authorities (Lazar et al., 2018). To understand such adaptation effectively, it is essential to view coastal systems through an SES lens. This approach enables us to examine how communities perceive, experience and respond to ongoing changes – the central focus of this study. Specifically, we aim to (1) identify how the Akplabanya community perceives changes in coastal SES, and (2) examine how the community is responding and adapting to those changes.

## Methodology

### Study area and people

Our study area is Akplabanya. It is a coastal fishing village near Ada, in the Greater Accra Region of Ghana. The people of Akplabanya are a subset of the Dangme ethnic group (Gyening, 1997). The Dangme people are an Indigenous group inhabiting the coastal area of the Greater Accra Region and part of the Eastern region of Ghana, located east of Accra. The Dangme people, particularly those in the inlands, such as Krobo, Sɛ and Osudoku, are traditionally known for their activities in farming, hunting, livestock rearing, trading and various native crafts and industries. These include the extraction of palm oil, blacksmithing, pottery and basket weaving, as well as the production of traditional soups, cosmetics and beads (Huber, 1993). Additionally, the coastal inhabitants of Dangme are recognized for their fishing expertise (Tanihu, 2017). Figure 1 shows our study location.

### Sample profiles

Based on our semi-structured interviews ($n = 61$), most people (89%, $n = 54$) were born in Akplabanya. The remaining 11% ($n = 7$) were born in other areas. Most participants (35%, $n = 21$) had lived in Akplabanya for over 56 years, while only a few (8%, $n = 5$) had lived there for <25 years. Most respondents (66%, $n = 40$) lived in southern Akplabanya, while 30% ($n = 18$) lived in western Akplabanya. A higher percentage of participants (41%, $n = 25$) had never attended school, while 36% ($n = 22$) had attained primary (grade school and middle school), 13% ($n = 8$) had attained secondary (high school) and 10% ($n = 6$) had attained tertiary education (undergraduate level). None of the respondents had a graduate-level education (master's level, PhD level and post-doctoral). Most participants interviewed were fishers (54%, $n = 33$), followed by fishmongers who combined other businesses with fish smoking during times of low fish catch (19%, $n = 12$). Other participants were solely fishmongers (17%, $n = 11$), while some relied on businesses related to the fishing livelihood in Akplabanya (5%, $n = 3$). There were also teachers (3%, $n = 2$). Only one participant mentioned being involved in canoe wood selling (2%, $n = 1$). All of the respondents engaged in some kind of livelihood activity. Regarding the range of monthly income, 41% of the respondents ($n = 25$) reported no fixed amount, followed by 21% ($n = 13$) who received 50–500 GH₵ (3.14–31.39 USD) monthly. In terms of family relocation, 83% ($n = 51$) of the families of the responders had moved out of Akplabanya for various reasons, while 10% ($n = 6$) had relocated due to climate-related stressors, such as recurring floods. Only 7% ($n = 4$) had families that had not moved out of Akplabanya.

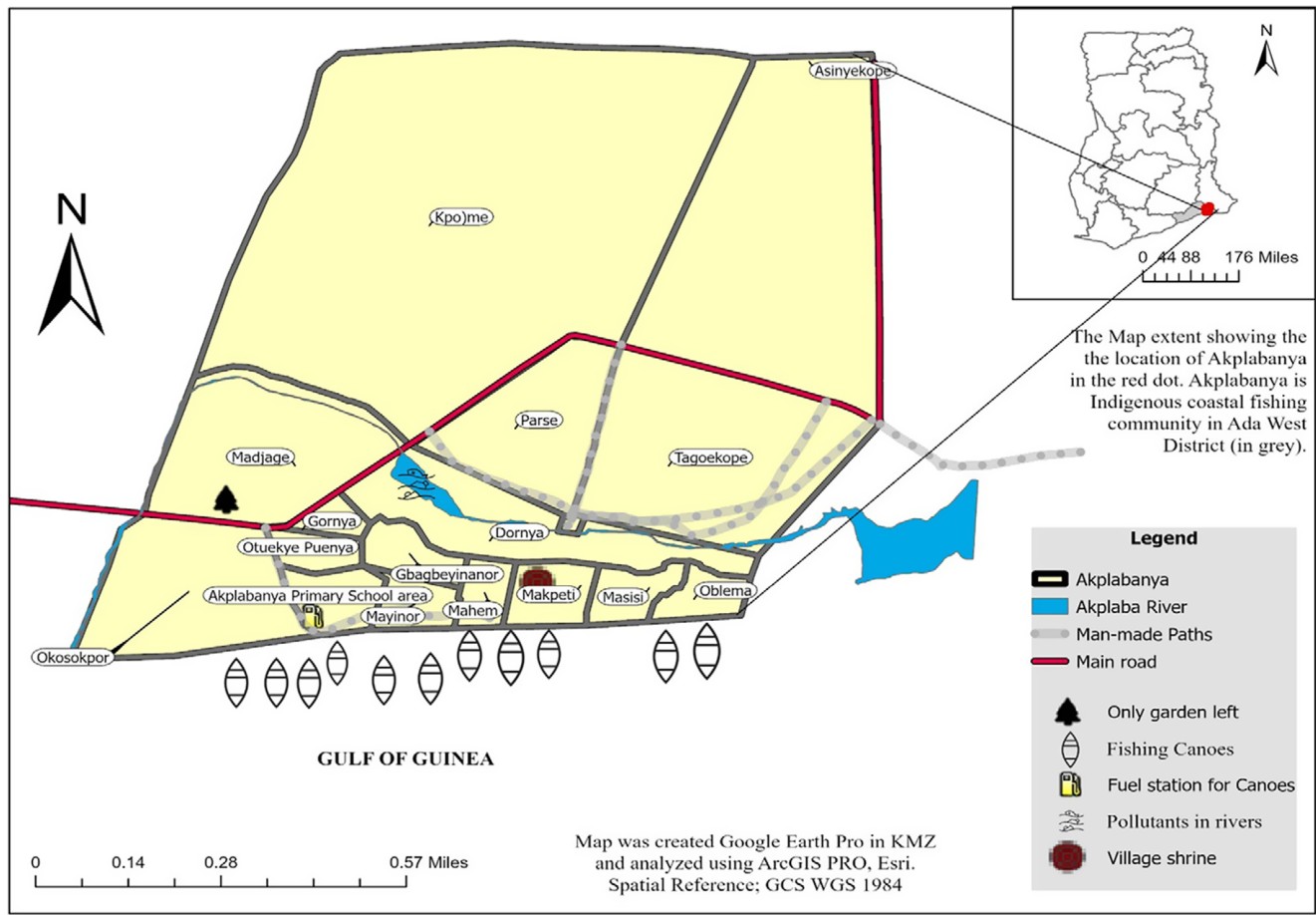

**Figure 1.** Map of Akplabanya and its extent in Ada West, Greater Accra, Ghana, West Africa. *Note:* This map represents the spatial context of Akplabanya in relation to the surrounding regions and its significance within the scope of our study. Akplabanya has 16 regions – namely, Oblema, Masisi, Mapketi, Mahem, Mayinor, Okosokpɔ, Akplabanya primary school area, Otuekye Puenya, Gornya, Gbagbeyinanor, Dornya, Madjage, Parse, Kpɔɔme, Asinyekope and Tagoekope, characterized by nucleated settlements.

### Theoretical approach

We took a SES approach to understand the interconnected yet distinct nature of the Dangme people and the Akplabanya subsystems (Berkes et al., 2003). The SES approach enables an understanding of how the Akplabanya community perceives changes in coastal SES by highlighting the interconnections between social and ecological components, rather than treating them in isolation (Folke, 2006).

### SES framework in understanding changes

A framework serves as a foundational structure that organizes key concepts and terms, guiding the way researchers describe, analyze and explore complex systems. Unlike a theory, it does not offer specific predictions but helps identify relevant factors and relationships for further investigation (Ostrom, 2005). The SES framework is intended to outline the key components and relationships that must be considered when analyzing SES, helping to focus on the essential elements and their interactions (Ostrom, 2009). Importantly, the SES framework supports the identification of both ecological and social changes over time, enabling researchers to better understand how communities respond to challenges and shifting system dynamics over time – which offers a conceptual foundation for examining how resilience emerges within the communities (Partelow, 2018).

### Resilience

Resilience refers to the ability to withstand change and continue evolving within constantly shifting environmental conditions (Folke et al., 2016). Within the context of SES, resilience is not about resisting change, but about maintaining essential functions and identities while navigating complexity, uncertainty and surprise (Walker et al., 2004; Folke et al., 2016). It emphasizes the intertwined nature of social and ecological processes and highlights the ability to learn, self-organize and innovate as key to sustaining development pathways in dynamic conditions (Berkes et al., 2003; Folke et al., 2005). As such, resilience thinking offers a powerful lens for understanding how communities respond to environmental challenges, adapt to stressors and, when necessary, shift toward entirely new trajectories of development (Olsson et al., 2006; Folke et al., 2016). In this study, we focus on enhancing resilience capacity – understood as the process through which communities develop the ability to absorb disturbances, adapt to change and retain their core functions and identity. We view enhancing resilience capacity as appropriate in the context of Indigenous communities like Akplabanya, where sustaining core cultural, ecological and livelihood functions is a critical goal.

*Characteristics of resilience.* According to Galappaththi et al. (2019, 2020, 2021), there are six characteristics of resilience by which community adaptation is assessed. These six characteristics

are place, agency, Indigenous knowledge, learning, collective action and institutions.

*Place*: The physical locations and social contexts that directly interact with and respond to climatic changes. It includes the geographical aspects, as well as the social structures, cultural values and community practices that support adaptive capacities and resilience building (Tuan, 1977; Galappaththi et al., 2019). Steel (1981) defines sense of place as a unique experience a person has in a specific setting. People have a strong emotional relationship with the place, which is known as place attachment. Studying place attachment integrates the physical, perceptual, psychological and socio-cultural dimensions of a place. Place attachment may therefore be influenced by various factors related to both the individual's experiences and the characteristics of the place itself (Najafi and Shariff, 2011).

*Agency*: The interaction between individuals and their environments, which includes individuals and communities to make decisions and take actions that enhance resilience and adaptive capacity to environmental changes (McLaughlin and Dietz, 2008; Parsell et al., 2017; Galappaththi et al., 2019; Ford et al., 2020). In other words, agency refers to the capacity of individuals and communities to make choices and take actions in response to environmental changes. This ability to act depends on several factors, including people's belief in their capability to manage and control events affecting them, as well as the socio-economic and political conditions that enable or constrain their actions (Ford et al., 2020).

*Indigenous and local knowledge systems*: Galappaththi et al., (2019; 20) has defined Indigenous and local knowledge as "the co-evolving cumulative body of knowledge (including observations, experience, lessons and skills) belonging to a specific human–environment system (or place) and handed down through generations by cultural transmission; reflects Indigenous and/or local people's cultural identity." Indigenous knowledge, particularly is defined as a cumulative body of wisdom and beliefs, passed down through generations via cultural transmission, concerning the relationships between living beings, including humans, and their environment (Berkes, 2003). Indigenous knowledge can be further referred to the understandings, skills and philosophies developed by societies with long histories of interaction with their natural surroundings, encompassing cultural practices, social interactions and environmental monitoring to manage and adapt to changes (Ford et al., 2020).

*Learning*: A behavioral change can be defined as learning (Barron et al., 2015). Learning can be further defined as the process of social learning, which involves collective action and reflection among various individuals and groups as they strive to enhance the management of interactions between humans and the environment (Galappaththi et al., 2019). Learning involves the ability to create, assimilate and process new information, evaluate different responses and reshape problems (Cinner and Barnes, 2019). Learning is central to Indigenous knowledge, characterized by experimentation, practice, regular interaction and openness to change. In Indigenous cultures, learning is characterized by ongoing and repeated interaction with environmental conditions. This approach fosters experience in handling these conditions and allows for effective responses through accumulated knowledge (Ford et al., 2020).

*Collective action*: Collective action refers to activities coordinated by a group of two or more people aiming to achieve a common objective (Galappaththi et al., 2019). It often involves coordination and cooperation among individuals to reach goals that would be difficult or impossible to achieve individually. Therefore, collective action involves more than one person and makes claims of agency

(or corporate) status (Olzak, 1989). Many Indigenous belief systems emphasize solidarity, communalism, loyalty and fellowship, which are reinforced through cultural practices and support collective action, thereby enhancing resilience in various ways (Ford et al., 2020). Collective can involve various social, political and economic contexts.

*Institutions*: Institutions are foundational structures in social life. They consist of established and prevalent social rules that guide interactions and behaviors. Institutions provide stability and predictability by creating consistent expectations of behavior, thereby enabling and constraining actions through formal and informal norms, such as laws, language and organizational systems (Hodgson, 2017). In other words, institutions are the established norms, rules and organizations, both formal and informal, that arise from social interactions and govern behaviors by determining what actions are required, allowed or prohibited (Galappaththi et al., 2019; Ford et al., 2020). In the context of Indigenous peoples, institutions are integral to Indigenous knowledge systems, deeply connected to specific places and culturally ingrained through practices, such as rituals, ceremonies, stories and other traditions. They are continuously created and reinforced within these cultural contexts (Berkes et al., 2000; Galappaththi et al., 2019; Ford et al., 2020)

### Data collection

We used a community-based participatory research approach to promote active community engagement. To achieve this, the author, Brandy Ayesu-Danso (BA-D), conducted 2 months of fieldwork in Akplabanya, guided and supervised by Eranga K. Galappaththi (EKG). During field data collection, the researcher received help from one language translator (Dangme to English) and one local research assistant. Before starting fieldwork in Akplabanya, we obtained Virginia Tech Institutional Review Board approval (No. 22-812). We followed the ethical guidelines outlined in the Institutional Review Board Protocol, which included obtaining informed consent from participants, ensuring confidentiality and respecting cultural norms and traditions (Babb et al., 2017). We ensured confidentiality and anonymity throughout our research. We were aware that the research was shaped significantly by the identities of both the researchers and participants, as individual backgrounds and characteristics influenced the entire process. As non-Indigenous researchers working on topics related to Indigenous peoples, we recognized that our positionality could meaningfully affect the research outcomes.

For primary data collection, we used a qualitative research design to understand how Akplabanya fishers experience and respond to SES change. Field data were collected using multiple methods: participant observation, semi-structured interviews, key informant interviews and focus group discussions. We have used four data collection methods to provide a rich, contextual understanding of the social-ecological changes and community adaptation strategies in Akplabanya. We used participant observations since they allow the researcher to gain first-hand insights into daily practices and interactions within the community. Participant observations helped us obtain contextual knowledge about Akplabanya fishers' experiences and responses to change. Our study involved observing various community activities in Akplabanya over 2 months. We used a field diary to record our daily observations. This included taking daily walks through the community, mapping the area, spending time with fishmongers and fishers at work, attending community events, meeting with the chief of

Akplabanya and the chief fisherman and observing fishers. We also took photos and videos to document the community's responses to environmental changes, such as sea-level rise, coastal erosion, storm surges and changes in precipitation and temperature patterns.

We used semi-structured interviews to explore individual experiences, perceptions of environmental and social change and personal adaptation responses in depth. We conducted 61 semi-structured interviews with fishers, fishmongers, fish traders and community committee members (Longhurst, 2003). We used a purposive sampling technique. The interviews were conducted for 1 month with the help of a predefined topic guide (Naz et al., 2022). The topic guide included questions such as how the Akplabanya people experienced climate change, how the Akplabanya people have experienced SES change and community perceptions about climate change (SM 01). The interviews varied in duration, lasting between 1 and 2.5 h, depending on the participant. We recorded the interviews in audio and video formats. Individuals who provided particularly rich or insightful responses during semi-structured interviews were later invited to participate as key informants or/and in focus group discussions. This approach was chosen to deepen our understanding of emerging themes and to allow for triangulation of perspectives of different data collection methods. We used key informant interviews targeting community leaders, elders and local experts to gain informed perspectives on various aspects, such as institutional arrangements, and collective strategies. Using topic guides (SM 02), we conducted 27 key informant interviews in Akplabanya (Lokot, 2021) to engage in in-depth conversations with community members who had specialized knowledge and expertise. We used 3 weeks of key informant interviews to track the history of perceived social-ecological changes and the history of adaptation responses to perceived changes among participants documented from semi-structured interviews. We used purposive sampling to recruit participants and interviewed the chief fisherman, heads of committees (Landing Beach Committee, Canoe Fishermen's Association and Premix Fuel Committee) and veteran fishers and fishmongers.

Later in the fieldwork, we conducted three focus group discussions to validate observations and responses from the participant observations, semi-structured interviews and key informant interviews, using topic guides (SM 03) developed from the collected data (Wilkinson, 2004; Onwuegbuzie et al., 2009). In each focus group discussion, we talked with 5 participants, for a total of 15 participants during our fieldwork. We employed purposive sampling techniques to recruit participants and ensure sample diversity in terms of age, gender and livelihood background. We recognized that power imbalances may have influenced the group dynamics and the willingness of some participants to speak freely – especially in mixed-age or mixed-gender settings. To mitigate this, we ensured a respectful and inclusive facilitation style, encouraged all voices equally and held discussions in familiar community spaces to create a comfortable environment. The first two focus group discussions were conducted to validate previously gathered data related to objectives one and two. Additionally, we aimed to assess the level of agreement among participants regarding the gathered data pertaining to objectives one and two. The third focus group discussion sought to validate the connections between perceived SES changes caused by climate change, responses to these changes adaptation structures, limitations of adaptation efforts and instances of maladaptation. Each discussion lasted between 2.5 and 3.5 h. The first focus group discussion was video-recorded, while the other two were audio-recorded with the consent of participants.

## Data analysis

Our study began with data analysis during the fieldwork stage, in which we simultaneously transcribed and coded using Microsoft Excel. We completed data transcription in 6 months. The data analysis process involved a systematic approach to identifying and organizing themes from the data collected from the semi-structured interviews, focus group discussions, participant observations and key informant interviews. Using thematic analysis, we analyzed the data to identify patterns and recurring ideas that represented significant aspects of the study's focus.

To achieve the first research objective, data were coded inductively. Inductive coding starts by carefully reading the text and reflecting on the different layers of meaning it may hold. The researcher then pinpoints specific segments of the text that convey significant ideas and assigns them to newly developed categories, each marked with a descriptive label (Thomas 2003). This process allowed the emergence of broad themes related to SES changes. Through iterative analysis, we built five themes by ourselves as coastal climate change, land use change, freshwater use, agro-biodiversity loss and pollution (Vaismoradi et al. 2016). Within each theme, subthemes were delineated to capture more specific aspects of the data. Themes and subthemes were categorized based on community perceptions of primary impacts and meanings (i.e., Voices from Akplabanya), rather than scientific or policy-centric definitions. For instance, we classified changes in the sea under "resource change" rather than "coastal climate change." This decision reflected the way participants framed these changes – not as abstract environmental phenomena, but as direct disruptions to resource availability and livelihoods. Since their interpretations centered on access to fish, erosion of fishing grounds and income loss, it was more accurate and meaningful to align these accounts with a resource-based framing. To quantify the prominence of each subtheme, we analyzed the frequency of mentions within participants' quotes. This involved systematically counting the number of times each subtheme was referenced across the dataset. These frequencies provided insights into the relative significance of each subtheme and highlighted areas of particular concern or interest as reported by participants.

To achieve the second research objective of identifying adaptation responses to perceived social-ecological changes, data were coded deductively. Deductive coding begins with pre-established theoretical concepts drawn from the literature, which are then systematically applied to the data to identify patterns that confirm, refine or challenge the theory (Pearse 2019). For this, we used pre-identified six main themes that were developed by Galappaththi et al. (2021) – place, agency, collective action, institutions, Indigenous knowledge and learning as an initial guide for the analysis. The collected data, consisting of quotes from participants, were systematically reviewed to determine whether they referenced the predefined themes. For quotes aligning with the main themes, we further analyzed the data inductively to identify subthemes, thereby allowing specific adaptation responses to emerge naturally from the data. This iterative process ensured that while the analysis was grounded in an established framework, it also incorporated participant-driven insights, thus capturing the details and unique context of the adaptation strategies. The identified subthemes provided a deeper understanding of the diverse ways in which communities respond to social-ecological changes. Figure 2 illustrates the themes and subthemes we identified, linking them to both study objectives, along with the data collection methods we used to develop each subtheme.

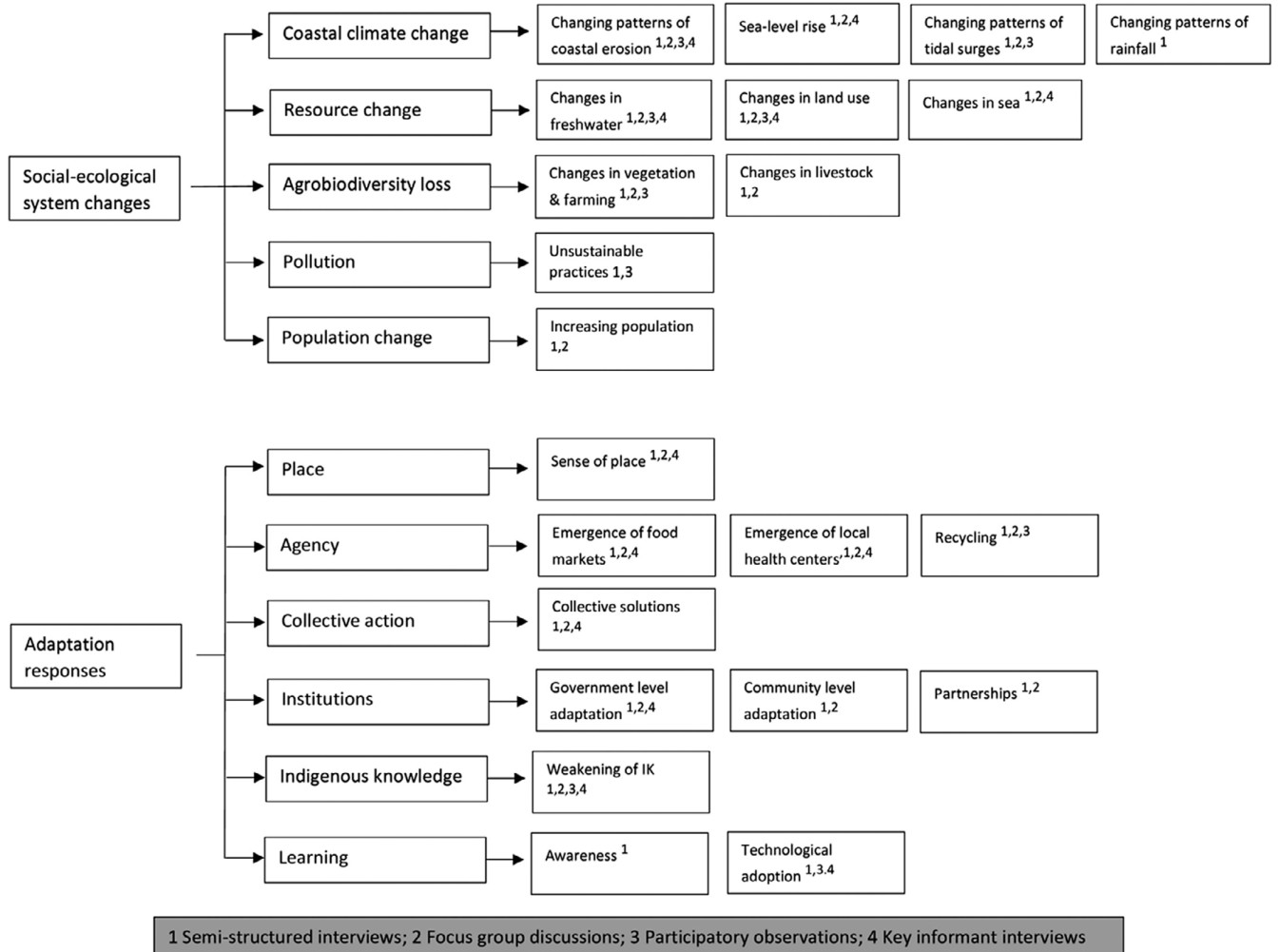

**Figure 2.** Data collection methods used in developing each theme and subtheme.

## Results

### Perceived SES changes

Our study identified four subthemes within the category of coastal climate change. The first subtheme was sea-level rise. Indigenous peoples in Akplabanya are experiencing a rise in sea levels (see Figure 3, Pictures A and B). One respondent said, "*I* (fisher) *observed that the sea level was farther away back then* (25–30 years ago)*, but it is much closer now*" (Semi-structured interview-SSI respondent 02). The constant tidal waves have led to the loss of fruit trees and grasslands. According to respondent 03 from SSI, "*Fruit trees like coconut trees no longer exist, grass used as a playground is also gone, and it all started after the continuous tidal waves.*" The second subtheme involved changing patterns of tidal/storm surges. Tidal surges have resulted in the loss of time, money and education while significantly affecting the community's children. SSI respondent 58 said, "*My boat, my fishing canoe or boat was scattered apart when we* (Indigenous people) *are flooded some time ago, in fact I* (fisher) *also lost an outboard motor. I've lost all these things.*" We identified changing patterns of coastal erosion as the third subtheme of coastal climate change. As shown in Figure 3, Picture A, the coastline of Akplabanya is eroding due to tidal surges that are carving away at the land. SSI respondent 27 said, "*The sand on the beach used to be plenty but the sea has taken it all back into the sea.*" Changing rainfall

patterns emerged as the fourth subtheme. The fishers of Akplabanya have lost their ability to predict fishing periods by observing rainfall patterns. Unexpected floods caused by rain prevent fishmongers from processing fish that the fishers catch. "*I* (fisher) *have experienced changes where seasons with fewer rainfall now have more and seasons with less sunshine are now warmer*" (SSI respondent 02).

The theme of resource change included three subthemes: changes in the sea, land use change and freshwater change. According to our study results, changes in the sea have resulted in alterations in fishing and the quantity of fish. Picture B illustrates this reduction, as the current fish catch is significantly lower than that of previous times (25–30 years ago). Fishers have relocated due to the decline in their livelihoods caused by the changing sea. While some fishers have returned, they face competition from large trawlers, which makes it difficult for them to catch certain fish species. SSI respondent 38 said, "*I* (fisher) *have moved, I travelled to Senegal in the year 2000. I went to sea (to fish there). I came back in the year 2013. I* (same fisher) *left for Senegal because the fishing here was low (at that time).*" Our study found changes in land use in Akplabanya, as participants described the loss of land, homes and infrastructure (e.g., electricity poles and water pipes). SSI respondent 61 said, "*We* (Indigenous people) *have a lot of land entering the sea. The land is also very salty that we cannot even grow anything there.*" Recurring floods caused by tidal surges have impacted the rivers, leading to the

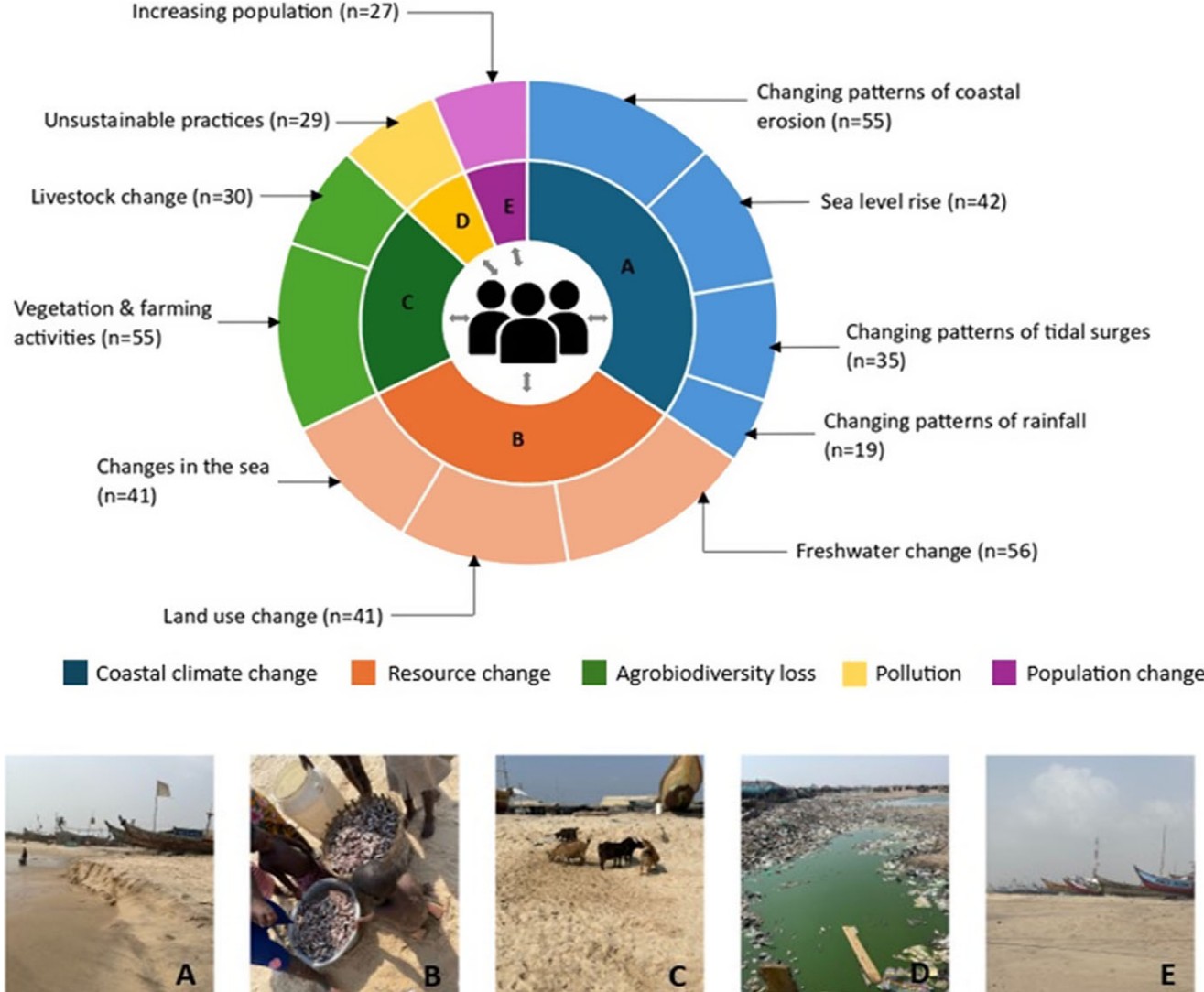

**Figure 3.** Identified themes and subthemes of social-ecological system changes in Akplabanya, along with photos. *Note*: The sunburst chart shows the perceived social-ecological system changes identified in Akplabanya. The inner circle displays the five main themes we identified. The outer circle shows the subcategories, along with the associated frequency of responses supporting each subtheme. The photos relate to the five main themes of socio-ecological system changes we identified: (A) coastal climate change, (B) resource change, (C) agrobiodiversity loss, (D) pollution and (E) population change. (Photos were taken during the fieldwork by BA-D.) This categorization is based on the community's framing of primary impacts. Overlaps between themes – such as resource change, coastal climate change and pollution – reflect the interconnected nature of SES dynamics and how respondents described and interpreted these changes. The two-way black arrows in the middle show the overlapping nature of the categories as perceived by community members.

loss of freshwater bodies. "*At the age of 18 years (about 27 years ago), Akplaba* (Akplaba River) *was so big that not everyone was allowed to go near it, but because the sea keeps pushing us back, and the consistent floods from tidal waves keep depositing sand in Akplaba, the river has lost its worth*" (SSI respondent 09).

Two subthemes were categorized under agrobiodiversity loss. They were vegetation and farming activities and livestock change. The recurring floods in Akplabanya, as reported by participants, have led to a decline in soil fertility. They have disrupted farming activities and resulted in the loss of both subsistence and commercial farm produce. This finding was supplemented by a quote from SSI respondent 16: "*Those days we* (Indigenous people) *don't buy food, we get food from our farms but today, farming has ceased completely in Akplabanya.*" Our study further revealed the loss of vegetation, such as coconut, neem trees and grasses. The loss of vegetation has resulted in a reduced quantity and quality of

livestock species. "*Animals are not able to get the grass field to feed on because the floods take away everything. So, you realize that the place becomes grassless. I* (fish monger) *had about 30 sheep, but today they are all dead and I have only two sheep left*" (SSI respondent 08). This was evident from our field photos, as Picture C shows livestock eating polythene and plastic instead of grass.

The subtheme, unsustainable practices, was categorized under the theme of pollution. Participants explained that they dumped refuse in the Akplaba River to make space for building due to the continual loss of land to the sea, as shown in Picture D. According to SSI respondent 09, "*Because the sea keeps pushing us forward and forward, we* (fish mongers) *are forced to dump refuse into the Akplaba River so we can make space to continue with our fish-smoking business.*" The theme of population change included one subtheme: increasing population. Our study found that the Indigenous population in Akplabanya has increased, causing a strain on

resources such as fish and land. "*With population growth, we* (Indigenous people) *have become so populated and I* (fisher) *realized that, increasing number of fishing boats and canoes on our shore here*" (SSI respondent 23). This increase was clearly visible on the coastline, as shown in Picture E.

### Key adaptation responses to perceived SES changes

The identified key adaptation responses to social-ecological system changes were assigned into six categories: place, agency, collective action, institutions, Indigenous knowledge and learning. We identified subthemes under each theme.

### Place

Indigenous identity are tied to place. The geographic location of a place determines access to natural resources, infrastructure and services that influence a community's ability to adapt to changes like climate variability or disasters (Lal et al., 2011). Our study found that Indigenous peoples in Akplabanya view "place" as integral to their identity and livelihood. Fishing is their primary connection to their "sense of place," which we categorized as the subtheme of place. Due to concerns about losing their ties to the sea, fishers in the community started using a technology known as light fishing, in which they employ light bulbs to attract fish with the aim of catching them. The use of light fishing technology has significantly increased the amount of fish caught in Akplabanya. As a result, more fishers who had previously left the community for better opportunities in other fishing areas returned. The use of light fishing technology shows how Akplabanya fishers' strong sense of place allows them to implement successful adaptation strategies and build resilience in their community.

> "*With this technology (light fishing), canoes of the various households have multiplied in Akplabanya, and some fishers have been able to see their (Akplabanya fishers') children through tertiary education.*" – SSI participant 19

The land is of secondary importance to the sea. Akplabanya's land is central to the fisher families. It serves as their residence and the place where children are educated. Here, fishers build, create and maintain their canoes, boats and fishing nets. Additionally, the land is the hub for distributing, processing and trading fish. Despite the loss of land due to tidal surges, sea-level rise and coastal erosion, some people of Akplabanya have remained in the community.

### Agency

Agency empowers individuals and communities to act on their unique knowledge, resources and connections to the environment, thus enabling them to adapt successfully to changing conditions (Fila et al., 2024). We identified three subthemes under agency: the emergence of food markets, local health centers and recycling. The loss of vegetation and farmlands has prompted significant changes in Akplabanya, including the development of small-scale supermarkets to meet community needs. Small-scale supermarkets have stepped in to provide coconut, cassava, okra, peppers and tilapia due to the scarcity of these farm produce items. This development highlights how agency enables Akplabanya communities to adapt creatively to environmental changes by leveraging available resources and knowledge in order to maintain access to essential goods.

> "*Formerly, there were some foods (from farmlands), today (2023) we (Indigenous people) don't have them (local food), but we buy them (food) from different places (small-scale supermarkets) and eat.*" – SSI participant 10

As an adaptive response to the loss of herbs like neem (*Azadirachta indica*), pharmacies and local health centers have emerged. We found that the community deals with the loss of vegetation caused by recurrent floods and coastal erosion by using barricades around small trees and tying bags full of used plastic bottles around the community. The plastic bottles are collected by individuals who sell them to recycling companies in exchange for money, which further helps reduce pollution.

### Collective action

By working together, communities strengthen their social and institutional networks, thereby improving their capacity to withstand and recover from climate impacts (Carmen et al., 2022). The Akplabanya community often faces tidal floods. Government agencies like the National Disaster Management Organization (NADMO) have been inconsistent in visiting the community during floods, thus leaving the community to fend for itself. Yet, Indigenous and local communities in Akplabanya have devised "collective solutions" as effective adaptation strategies. This served as a subtheme under the main theme. As an adaptation response to environmental challenges, community members have created water channels to redirect floodwater back to either the sea or the Songor Lagoon. This proactive measure provides the Akplabanya people with some level of comfort regarding the resumption of their daily activities despite the recurring floods.

> "*Anytime we (Akplabanya people) are flooded, we have to create water channels and dig gutters for the waters to flow back, that's the only thing that we do before we can become free.*" – SSI participant 21

### Institutions

By establishing clear rules and norms, institutions create predictable environments that guide behavior during uncertainty (Wallis, 2022). Akplabanya fishers show multilevel institutional adaptation responses, for which we identified three subthemes: government-level adaptation, community-level adaptation and partnerships. We found that the government has implemented adaptation measures in Akplabanya. The government of Ghana, through the Fisheries Commission, implemented a 2-month national fishing holiday in 2019 to allow for fish replenishment. This policy was a response to low fish catch along the coast of Ghana. It has been found that the policy led to an increase in fish catch in Akplabanya, thus improving the community's resilience to fish catch-related issues. The NADMO in Akplabanya is responsible for managing natural disasters, such as floods from tidal surges, in Ghana. It provides the community with basic items, such as rice, mattresses, and used clothing, for the Unit Committee to distribute among households. This coping strategy by NADMO is its response to the loss of basic items during floods caused by tidal surges, and it helps reduce the vulnerability of affected households.

However, our study uncovered an inconsistency in Akplabanya's government support. We found that the people of Akplabanya received government assistance for dealing with floods from tidal surges in 1997 and 2013. The District Chief Executive oversaw the projects. The first project – a small sea defense structure built in 1997 – was destroyed by tidal surges within a year. The second project, started in 2013, is a sea defense system that is still under construction and has not yet reached the shores of Akplabanya. Both projects were suspended due to the government's shifting environment.

> "*About two decades ago, we (Indigenous people) cried out to the government (District Chief Executive of Ada West) at that time to*

*come to our aid concerning the recurrent floods from tidal surges, so the government brought in huge blocks of stones and deposited them at the shoreline but [this] got destroyed after a year. The project was incomplete because the government changed power and the focus on the community shifted. After that, about a decade ago (2013), the government started building sea defense from Ada Foah, the capital town of all the Ada Districts to Totope, a village along the coast but the project stopped when power changed hands again."* – Focus group discussion 2

Community adaptation outperformed failed government efforts. Our study revealed that when government agencies failed to provide assistance, the village chief and his cabinet organized the community water channels at both the east and west ends of the community. This action was an institutional response to cope with floods caused by tidal surges that mainly affect the community at dawn. We found that local institutions are partnering with the Landing Beach Committee, Canoe Fishermen's Association, and Premix Fuel Committee to raise funds to assist fishermen affected by strong tidal surges. These funds were used to support fishers whose canoes, fishing nets, outboard motors, boats, generators and so forth, have been destroyed during strong tidal surges. Although this response from local institutions is reactive, it helps fishers recover their livelihoods and makes them more resilient.

### Indigenous knowledge

Rooted in local experience, Indigenous knowledge provides contextually appropriate solutions to environmental and social challenges, which is important in building community resilience (Galappaththi et al., 2021). In Akplabanya, we noticed a "weakening of Indigenous knowledge," which we categorized as a subtheme. The community has shifted from traditional herbal medicine to local remedies and modern healthcare. This change is a response to the depletion of vegetation and the loss of the older generation, which held the Indigenous knowledge of traditional herbal medicine. The transformation is evidenced by the fact that fewer households are growing important herbs for local remedies. Consequently, visits to hospitals and clinics for modern healthcare have increased.

*"Back then (25–30 years ago) fresh leaves of the neem tree (Azadirachta indica) were used to manage and cure fever."* – KII participant 28

We discovered that 25–30 years ago, the community had Indigenous knowledge in terms of constructing houses with clay soil and thatch. However, due to recurring floods caused by tidal surges, this knowledge has gradually been replaced by the use of building blocks, which offer greater durability against flooding. Most households now live in block structures. While block buildings are less prone to collapse during floods compared to clay and thatch, they are constructed using beach sand, which has led to significant wear and tear over time. In response, the community has adopted coping strategies, such as replastering and painting, to maintain the integrity of their homes. This shift represents yet another instance of the gradual loss of Indigenous knowledge in favor of more modern but not entirely problem-free construction practices.

### Learning

The process of learning enables individuals, communities and systems to acquire, process and apply knowledge to respond to changing circumstances and future uncertainties (Fullerton et al., 2021). We identified two subthemes under this category: awareness and the use of technology. The Akplabanya community is aware of the concept of climate change. This is because they have had meetings with government agencies like NADMO. We found that community awareness is a response to issues, such as sea-level rise, frequent flooding and coastal erosion, due to the changing environment along the coast of Akplabanya. This awareness can create learning opportunities with regard to local adaptation responses, such as the burying of pollutants at the beach. Furthermore, we observed that fishers in Akplabanya use mobile phones to share information about fishing among themselves. Additionally, they have purchased and learned to use Global Positioning System (GPS) devices to aid in navigation at night. The acquisition of such knowledge is the fishers' response to low fish catch, and it enhances the community's resilience.

*"So, we (one of the fishing groups among Indigenous people) were the first people to get GPS, which ha[s] been giv[ing] us directions. And sometimes when we have fog on the sea, it gives us directions to where we are going, and then sometimes it direct[s] us to where the fishes are, and then we go and cast it there and we catch fish, so I realize that it was we who brought it, and we realize that it is helping us so people are now buying it."* – SSI respondent 61

## Discussion

Our study contributes to the growing literature on coastal SES by documenting changes in Akplabanya, a coastal Indigenous community in Ghana. Coastal communities worldwide are vulnerable to climate change and environmental degradation, but the specific experiences and adaptive strategies of communities like Akplabanya remain underexplored (Berkes and Jolly, 2002; Adger et al., 2005). The study illustrates five key themes of social-ecological change (i.e., stressors and shocks): coastal climate change, resource change, agrobiodiversity loss, pollution and population change. Specific changes identified under these themes – which include sea-level rise, coastal erosion and rainfall patterns – have altered the physical environment, while social factors, such as increasing unsustainable practices and increasing human population, further complicate the challenges facing coastal systems. These findings align with global observations on the interconnectedness between environmental and societal changes in coastal areas (e.g., Cinner and Barnes, 2019; Kanan and Giupponi, 2024).

While climate change is a global phenomenon, we used the term "coastal climate change" to refer to the localized manifestations of climate-related impacts as experienced in coastal areas. This includes sea-level rise, coastal erosion and saltwater intrusion, all of which disproportionately affect coastal communities (Nicholls and Cazenave, 2010). Although climate change cannot be geographically limited, the experiences and consequences of climate change are often place-specific, and in our case, these impacts were observed and described within the context of Akplabanya's coastal environment. Thus, the theme "coastal climate change" captures the community's place-based framing of environmental changes (Adger et al., 2013).

Localized strategies play a significant role in addressing climate-induced changes in Akplabanya. For example, using light fishing technology and constructing block buildings are innovative responses grounded in the community's understanding of their environment. These adaptations showcase how Indigenous knowledge and agency are critical for building resilience (Ford et al., 2020). However, these responses raise questions about long-term sustainability, such as the reliance on beach sand for construction. Thus, they highlight some vulnerabilities. Studies from other coastal regions have demonstrated

the need for integrating local adaptations with external support to ensure the resilience and sustainability of adaptive strategies (Berkes et al., 2003; Refulio-Coronado et al., 2021). For instance, in the Philippines, partnerships with NGOs have provided technical expertise and funding for mangrove replanting projects, thus complementing local efforts to combat the effects of rising sea levels (Walters, 2004). In Indonesia, community-based initiatives have collaborated with international organizations to implement sustainable aquaculture practices and restore degraded coastal ecosystems, such as coral reefs and mangroves, with organizations training community members on how to restore coral reefs (*United Nations*, n.d.). In Akplabanya, combining such external support with local adaptation efforts could strengthen the community's capacity to adapt and thrive.

Collective action emerged as a cornerstone of adaptive strategies in Akplabanya. The community creates water channels to manage flooding and organize community responses during disasters; these collaborative efforts illustrate the power of social cohesion in building resilience (Carmen et al., 2022; Dharmasiri et al., 2025). Akplabanya also exemplifies the critical role of social capital in climate adaptation (Adger et al., 2005; Folke, 2006). This mirrors the collective action taken by communities in Bangladesh, where the regular threat of flooding has prompted locals to organize, build and maintain embankments and levees, thereby demonstrating the power of community-driven adaptation (Ensor and Berger, 2009; Dewan, 2022). Furthermore, local institutions have played a supportive role by assisting fishers during tidal surges triggered by strong storms resulting from climate change. Though local institutions have been supportive in Akplabanya, their efforts do not mirror the actions of community-based organizations in the Philippines. Local institutions in the Philippines have established early warning systems and disaster preparedness plans to protect fishermen and coastal communities from typhoons and extreme weather events (Lassa et al., 2019). Social capital plays a vital role in enabling these adaptive efforts, as it fosters cooperation, mutual support and effective communication within the community. It facilitates the pooling of resources, the coordination of collective actions and the establishment of strong networks of trust and shared responsibility; these not only enhance the community's capacity to respond to immediate climate-related challenges but also strengthen long-term resilience and adaptive capacity (Pelling and High, 2005; Adger, 2010).

Our study highlights a concerning trend: the weakening of Indigenous knowledge systems. In Akplabanya, the community shift from traditional herbal medicine to modern healthcare reflects broader patterns of cultural loss linked to environmental degradation and generational shifts (Berkes et al., 2000; McGregor, 2004). For example, younger generations are increasingly unfamiliar with local plant-based healing practices that older community members once widely used. While modern healthcare showcases the resilience of the healthcare system (Anderson et al., 2020), it also limits reliance on locally rooted practices such as harvesting medicinal plants and applying Indigenous and local knowledge. The weakening of traditional knowledge systems is also documented across the globe, including among Inuit in the Arctic (Galappaththi et al., 2019) and Sri Lankan Vedda (Galappaththi et al., 2020). Among the Inuit, younger generations are increasingly disconnected from traditional practices, such as sea ice navigation, hunting techniques and weather prediction, due in part to climate change and the influence of modern technologies. Similarly, the Vedda have experienced a decline in forest-based knowledge, including hunting, foraging and medicinal plant use, as a result of resettlement policies, deforestation and integration into mainstream society. As Galappaththi et al. (2021) recorded, documenting and integrating such knowledge into broader

adaptation strategies (e.g., National Adaptation Plan) could preserve cultural heritage while enhancing community resilience.

Formal governance plays a significant role in addressing social-ecological changes. Institutional responses in Akplabanya, including disaster management efforts and fishing holidays, show strengths in supporting community resilience. Yet, the inconsistent implementation of government-led projects, such as the sea defense structure, indicates systemic challenges in sustaining institutional interventions (Abugre, 2018). The case of Akplabanya can benefit from multilevel governance frameworks that integrate community-based and governmental efforts, which are strongly supported by studies on adaptive co-management (Olsson et al., 2007; Bodin, 2017). For example, adaptive co-management is widely used in the Canadian Arctic to manage fisheries; both community and government institutions are the co-managers of the local fisheries, which involves collaborative decision-making and sharing authority (Galappaththi et al., 2019).

Learning and using technology emerged as an adaptive response in Akplabanya. Technologies, such as GPS devices, reached the community later than the rest of the world, and they improved daily fishing practices. Such adaptive learning reflects the community's capacity to innovate in response to environmental and economic challenges (Cinner and Barnes, 2019; Fullerton et al., 2021). External technological interventions, such as government and nongovernmental organizations, could further advance adaptive capacities by enhancing access to technology and training, particularly if paired with community-led initiatives and knowledge-sharing networks. Similar findings from other coastal regions highlight the transformative potential of integrating traditional practices with modern tools (Ford et al., 2020). For example, in Vancouver, Canada, an interactive online map has been developed to project future sea-level rise along the North Shore. This tool enables local communities to visualize and understand potential coastal changes, thereby aiding in effective adaptation planning and raising awareness about climate change impacts (DiPaola et al., 2023).

One limitation of this study is that the identified themes are not always mutually exclusive; many naturally overlap, reflecting the interconnected nature of social-ecological changes in the community. For instance, the boundaries between resource change and biodiversity loss are often blurred. This underscores the inherent complexity of SES, where changes cascade across ecological, economic and social domains. Rather than imposing artificial separations, our approach centers community framings, recognizing that overlapping sub-themes are a characteristic – not a flaw – of SES dynamics (Berkes et al., 2003). Future research could employ multi-coding or integrative thematic analysis to more clearly trace these overlaps and their implications. Another limitation of this study is that the adaptation responses we identified – such as the adoption of light fishing – may also constitute maladaptation, depending on contextual drivers like geographical location and resource governance. While light fishing in Akplabanya appears to enhance short-term resilience and reinforce a strong sense of place tied to fishing (Lal et al., 2012), similar technological adaptations have been shown to lead to resource overexploitation and long-term ecological harm in other contexts (Barnett and O'Neill, 2010; Eriksen et al., 2015). Thus, what appears adaptive for one community may have unintended negative consequences elsewhere, highlighting the importance of evaluating adaptation within specific socio-ecological settings.

Drawing from our study findings, we have identified five key lessons and actionable insights to enhance resilience in coastal communities. They are as follows: (1) Climate change impacts, such as sea-level rise and agrobiodiversity loss, are compounded

by human-induced pressures like population growth and unsustainable practices (*interconnected nature of social-ecological changes*); (2) integrating local efforts with external technological and institutional interventions will enhance the resilience of coastal communities (*need for external support and multilevel governance*); (3) social capital proved essential for resilience, enabling collaboration, resource sharing and strong community networks (*role of social capital*); (4) it is important to document and integrate traditional knowledge into formal adaptation strategies that can preserve cultural heritage while strengthening adaptive capacities (*preservation of Indigenous knowledge*) and (5) the introduction of technologies, such as GPS devices, demonstrates the capacity for adaptive learning and innovation within the community (*technological innovation and learning*).

## Conclusions

Our study focused on understanding social-ecological changes and the adaptation responses of the Akplabanya fishing community. We identified coastal climate change as the most prominent social-ecological change that community members experience. Despite the severe impacts on the coastal area, the strong sense of place among Akplabanya fishers drives them to adopt diverse adaptation strategies while remaining in their community. This highlights the critical need to enhance their resilience and thereby ensure long-term sustainability. Our study offers key lessons and actionable insights that can guide resilience-building efforts in Akplabanya and other coastal communities facing similar challenges.

**Open peer review.** To view the open peer review materials for this article, please visit http://doi.org/10.1017/cft.2025.10011.

**Supplementary material.** The supplementary material for this article can be found at http://doi.org/10.1017/cft.2025.10011.

**Data availability statement.** The data will be made available upon request at any time.

**Author contribution.** Conceptualization: E.K.G., B.A-D. and S.M.J. Funding acquisition: E.K.G. Investigation: E.K.G. Methodology: E.K.G., B.A-D. and S.M.J. Field data collection: B.A-D. Supervision: E.K.G. Writing: S.M.J., E.K.G. and B.A-D. Writing – review and editing: E.K.G., T.D.B., A.B. and S.R. Visualization: S.M.J., E.K.G. and B.A-D. Project administration: E.K.G.

**Financial support.** The authors sincerely acknowledge the funding support received by the Department of Geography in the College of Natural Resources and Environment (CNRE) at Virginia Polytechnic Institute and State University.

**Competing interests.** The authors declare none.

**Ethics statement.** This study received approval from the Virginia Tech Institutional Review Board (IRB No. 22–812). Informed consent was obtained from all participants before their inclusion in the study.

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
