## [Reviewer Report]

The manuscript explores the socio-ecological changes in the coastal community of Akplabanya, Ghana, focusing on the community’s perceptions and adaptation strategies. Using a qualitative approach, the authors identify key themes of social-ecological change, such as coastal climate change, resource changes, biodiversity loss, pollution, and population change. The study applies a Social-Ecological Systems (SES) framework and emphasizes the role of Indigenous and local knowledge in adaptation strategies.

The subject is very interesting and definitely important to address the knowledge gaps relevant to the Indigenous/local community’s perceptions of change and their adaptation responses in remote/rural communities.

While the manuscript has the potential of being published, in this format, this manuscript looks like a first draft of a PhD thesis chapter. Most importantly, the manuscript failed to adopt valid/suitable methodological approaches hence, providing rich and reliable findings/conclusions. I think the methodology section is the most important part of each research as it determines the validity and reliability of data.

Apologetically, I have to say that, the research methodology did not seem robust enough (or was not taken as seriously as it should have been) and the authors used some of the trending concepts (such as resilience, SES approach, etc.) as buzzwords without justifying how/why they were recruited. The application of those terms and concepts was very generic and, and their adoption was not backed up by using highly important citations/references in the scholarship.

Introduction:

Needs more detailed development of the problem, how this paper is going to address the problem and extend the existing knowledge in the scholarship.

Aims and objectives: A little clearer wording makes it a lot more understandable: “Specifically, we aim to (1) identify changes in coastal human-environment interactions as perceived by the Akplabanya Indigenous community and (2) examine the Akplabanya community’s human adaptation responses to those changes in the coastal human-environment interactions.” For example, explain what is being examined exactly.

Methods:

Unfortunately, the method section is not well developed and there is significant structural and application confusion that needs to be addressed:

Differentiating the differences between “research methodology” and “research methods”. The section is called Research Methods, and it addresses the broader viewpoints/approaches to the research not only the methods. So, I think the title should be Methodology rather than Methods.

L8: “we used a qualitative research design …” A qualitative research design applies to the design of the entire research. Perhaps the authors meant qualitative research methods for data collection, not the design of the entire research (as mentioned above)

The authors refer to four research methods without any further explanation of why they selected these methods in the beginning and what types of data they projected to collect/generate by each method.

While the authors indicate “participant observation” as a method used in their research, a few lines after they address “participatory observation”. Are these two methods the same and are they used interchangeably? or they are different?

The authors indicate that they have used snowballing and random sampling for both semi-structured and key informant interviews. Due to the differences in the application of the two methods, don’t they think that the sampling methods should be different or more purposeful for key informant methods? How the authors can make sure that they reach out to the key-informant interviewee by random sampling?

P7, L47: “We employed purposive sampling techniques to recruit participants, ensure sample diversity, and address power imbalances”. What is a power imbalance?

Requires more explanation as to why the focus on explaining the themes was on climate change while in the abstract and methodology section, they addressed other types of changes.

P8, L18: “To achieve the first research objective, data were coded inductively”. What does it mean to “code data inductively”? And the same for the “coding deductively”.

Section 3.2. The sample profile is better fitted in the methodology section than the results. It does not seem to be the expected finding of the research.

Section 3.2 repeats the information addressed earlier which makes the manuscript unnecessarily verbose

I think the authors should explain/elaborate more on the theoretical approaches they adopted and explain why they selected the SES approach, those six themes and how they think these approaches assist with their analysis. For example, they should explain if the 6 themes were adopted from other references, or if they came up with these six themes according to participants' responses. I do not think just a short paragraph is enough as these approaches play an important role in the research. For example, SES approach is more than just a generic term to be used as a buzzword in research, it has a strong conceptual background, hence calling it an approach. The authors need to convince the readers that they understand the SES approach and justify how they used this approach to develop a valid conclusion.

The manuscript is verbose in a lot of sections and there are lots of repetitions and unnecessary explanations that could be removed (e.g. the six themes were repeated in a couple of places which seems unnecessary) and more useful information gets incorporated.

Results:

Section 3.2. Fails to provide a clear understanding of SES drivers of change.

For example, the authors identify 5 main themes with several sub-themes. It is not clear how these five themes were connected, or should be related to the other 6 themes addressed before (both called themes).

The labelling and allocation of sub-themes within the main themes were a bit confusing and unconvincing as well. For example, one theme for drivers of change is labelled as coastal climate change: what does coastal climate change mean? Can you limit climate change to just coastal areas?

Also, the author should explain several overlaps in categorising the themes and sub-themes. For example, the loss of ocean fish stock seems to be related to biodiversity loss. So why biodiversity change was not merged with resource changes? In addition, the authors addressed farm activity and livestock changes within the biodiversity changes, which is a little unusual categorisation. I think the whole section needs to be restructured and rewritten

More explanation is needed regarding the similarities and close relation between themes: learning and knowledge, agency and collective actions. Etc.

Some of the adaptation responses (such as light fishing) seem to be maladaptation strategies rather than adaptation.

I am not convinced what was indicated under the Agency theme is actually adaptation responses (forming local markets, health centres etc.)

How collective actions are different from agency? And institution different from all of them?

General, editorial and semantic comments:

There were multiple addresses to enhancing resilience or resilience building in the manuscript. The newer approaches to SES resilience do not always suggest enhancing resilience, they mostly focus on “ managing resilience” which could consider eroding resilience to facilitate a deliberate SES transformation. An explanation of why the authors focus on enhancing resilience could be useful.

P10 L19. The authors indicated that they applied an SES approach but the language/terms they used do not align with SES discourse. For example, the authors said: “We identified changes in the social-ecological system”. From an SES approach point of view, “change” should be “drivers of change” and it is easier to grasp what the author means when they just say “changes”

Consistency in tenses: either simple past or simple present

- the whole manuscript needs to be revised and edited for the English language and the Clarity of concepts/sentences, such as:

“Coastal ecosystems and social systems are greatly interconnected, with significant climatic and non-climatic changes occurring over the past decades (P3, L19).

Coastal SES studies emphasize the evolving interactions between social and ecological factors, thus pointing to the risks that environmental and social changes pose with regard to resilience (P3, L26)

sea erosion? (p3, L42)

What does it mean “one-meter rise”? “As projected by Climate Central’s coastal risk screening tool, Akplabanya faces the looming threat of complete submersion under one meter of water by 2030”.

Needs cleaner and clearer wording also needs references: “The SES approach helps identify changes in coastal human-environment interactions as perceived by the Akplabanya community, emphasizing neither ecosystems nor societies inisolation but, rather, the connections between the system’s ecological and social components.”

“Note: This map represents …” is the note part of the map caption or main body? Or should be put as a footnote?

P6, L47) “To achieve this, the author…” should be authors? please check for other grammatical issues.

---

## [Reviewer Report]

Review of manuscript:” Voices from Akplabanya: Community Adaptation and Social-Ecological Changes in Coastal Ghana”

Authors: Galappaththi, Ayesu-Danso, Jayasekara, Baird, Bukvic, Rijal

Manuscript ID: CFT-2025-0002

This paper is timely and presents a valuable opportunity to highlight the experiences of communities like Akplabanya, which inherently and collectively respond to environmental pressures. The manuscript also emphasises how these local adaptations have a short term effect and underscores the pressing need for external, coordinated, technical, and expert support, as well as local institutional consistent interventions to ensure long-term adaptive sustainability resilience. Overall, the paper offers actionable reflections relevant for rural coastal communities facing similar social-ecological challenges.

The introduction sets up a strong conceptual framing, however, the first portion reads somewhat detached and could benefit from stylistic improvements to enhance flow and readability. A reordering of certain paragraphs may help achieve this. For example, I would move P3 L 19-32 before P3 L3.

There is some ambiguity to what the term “its” in P3 L34 refers to, and the authors should clarify this to avoid misinterpretation.

I would include the reference to t the figure just after mentioning the focus of the coastal community (P4 L19) to avoid the sentence in P4 L28.

The methodology section would benefit from a more detailed explanation of the Social-Ecological Systems (SES) approach, especially in lines 50–51 on page 4. The current transition from the broad SES framework to the six selected themes is abrupt. A short expansion on the cited Galappaththi literature and how it informed the selected themes would provide better continuity and methodological clarity.

The reference to authors who conducted the fieldwork and guided should be written in full the first time it is listed (P7 L47-48).

The authors should also clarify whether the individuals who participated in semi-structured interviews, key informant interviews, and group discussions were distinct across these categories or if there was overlap. If participants were involved in multiple aspects of data collection, this could raise questions about the independence of responses and introduce a degree of subjectivity into the analysis.

The difference between “tertiary education” and “graduate-level education” should be explained more clearly (P 9, L55), as the current wording suggests a contradiction or inconsistency in the reported educational levels.

In the results section, while references to photographs help add contextual richness, their evidentiary function should be treated with caution. For example, on P 12, L29–30, the sentence “Picture B illustrates this reduction, as the current fish catch is significantly lower than that of previous times” implies a causal visual representation that may not be possible. A photograph alone cannot empirically demonstrate a reduction in fish catch over time, so the wording should be moderated accordingly.

Finally, in the discussion (page 17, line 12), the argument would benefit from a clearer connection between global-scale observations and the specific realities of rural, localised adaptation. Expanding this sentence to bridge the gap between the broader climate discourse and the grounded, community-level responses in Akplabanya would reinforce the significance of localised strategies, and provide a stronger lead-in to the subsequent discussion of adaptation practices.

---

## [Editor Report]

Both reviewers recognize the valuable contribution of your manuscript in addressing knowledge gaps relevant to the Indigenous/local community’s perceptions of change and their adaptation responses. They did, however, indicate major revision is necessary particularly to the writing style, clarity and length of the manuscript.

---

## [Reviewer Report]

Thanks very much for reviewing the article and applying my comments. The manuscript has undergone substantial revisions and shows significant improvement from what appears to have been an earlier draft. Many of the previous comments regarding structure, clarity, and methodological justification have been addressed. The authors have clearly put effort into refining the language and integrating the theoretical framework more explicitly.

While considerable progress has been made, there are still areas where further refinement would enhance the manuscript’s overall quality and impact.

1. English Language and Clarity: There are definite improvements in clarity, but some sentences still require editing for grammatical correctness, conciseness, and flow. For example, the manuscript still exhibits some inconsistencies in tenses, frequently shifting between past and present tense within paragraphs or even sentences. For example, in the abstract: “We found social-ecological changes related to five themes: … As adaptation responses to these changes, the community adaptive responses we found are [were?] a) place...”. The shift from “found” (past) to “are” (present) is an example. A consistent past tense for describing the study’s findings and actions (e.g., “we found,” “we identified,” “the community adopted”) and present tense for general facts or framework definitions would improve clarity.

2. Labelling, Allocation of Sub-themes and Overlaps in Categorisation (important note) : While there’s a sunburst chart (Figure 3) which attempts to visualise themes and sub-themes, some overlaps and unusual categorisations still exist and could be further refined for maximum clarity. While the discussion attempts to explain the interconnectedness of themes, some specific categorisations still seem slightly off. For instance, “loss of ocean fish stock” is a sub-theme under “Changes in the sea” (which is under “Resource change”), and “livestock change” is under “Biodiversity loss”. I previously asked why biodiversity change wasn’t merged with resource changes if ocean fish stock loss relates to biodiversity. The current structure separates them (Resource Change and Biodiversity Loss are distinct main themes). While there’s a logical connection, the authors seem to have categorised based on the primary impact or perception. For example, “Changes in the sea” (resource change) might be about availability of fish, while “Biodiversity loss” is about species loss or ecosystem health. This distinction could be made more explicit. Placing “farm activity and livestock changes” under “biodiversity changes” is still an “unusual categorisation” as farm activities are typically land-use or economic changes, and livestock changes could be resource or economic, rather than biodiversity. If the loss of vegetation due to climate impacts, then affects livestock, it could still be framed more clearly.

The authors say that” We acknowledge the overlaps among themes and sub-themes, particularly between biodiversity change and resource change. These overlaps reflect the complex and interconnected nature of SES dynamics. We have added this important point to our discussion as a limitation of the study as follows: One limitation of this study is that the identified themes are not always mutually exclusive; many of them naturally overlap, reflecting the interconnected nature of social-ecological changes in the community (J. Berkes et al., 2003)

I am not sure if that is an inherent complexity due to the complex nature of SESs. For example, I do not think farm animals (livestock) are considered as biodiversity in any valid references.

3. I recommend all the terms addressing “human-environment interaction” be changed to SES. SES is not just a new term to indicate human-environment interactions. SES is backed up by a particular understanding of the core notions of understanding of resilience, adaptation and transformation.

Thanks very much again for your time and effort, and I hope to see this paper published after the revision.

---

## [Editor Report]

Although your revisions have substantially improved the manuscript, some minor edits are still needed.